# A Rewriting Approach for Gender Inclusivity in Portuguese

**Leonor Veloso** and **Luísa Coheur** and **Rui Ribeiro**
INESC-ID, Lisbon, Portugal
Instituto Superior Técnico, University of Lisbon, Portugal
leonor.veloso@tecnico.ulisboa.pt
luisa.coheur@tecnico.ulisboa.pt
rui.m.ribeiro@tecnico.ulisboa.pt

## Abstract

In recent years, there has been a notable rise in research interest regarding the integration of gender-inclusive and gender-neutral language in natural language processing models. A specific area of focus that has gained practical and academic significant interest is *gender-neutral rewriting*, which involves converting binary-gendered text to its gender-neutral counterpart. However, current approaches to gender-neutral rewriting for gendered languages tend to rely on large datasets, which may not be an option for languages with fewer resources, such as Portuguese. In this paper, we present a rule-based and a neural-based tool for gender-neutral rewriting for Portuguese, a heavily gendered Romance language whose morphology creates different challenges from the ones tackled by other gender-neutral rewriters. Our neural approach relies on fine-tuning large multilingual machine translation models on examples generated by the rule-based model. We evaluate both models on texts from different sources and contexts. We provide the first Portuguese dataset explicitly containing gender-neutral language and neopronouns, as well as a manually annotated golden collection of 500 sentences that allows for evaluation of future work.

## 1 Introduction

The relationship between language and gender, as well as its effects on societal gender dynamics, has been examined and documented as early as the 70s (Gal, 1989, 1978). In more recent years, there has been a push towards the usage of gender-neutral (or *gender-fair*) language. How this gender-fairness is achieved, is, however, highly dependent on the specifics of each language. During the course of this work, we will concern ourselves with the case of Portuguese.

Like most Romance languages, Portuguese is characterized by a binary grammatical gender system in which nouns belong to one of two classes: masculine or feminine. During the last decade, with the advent of social media, there has been an increase in the visibility and usage of new sets of gender-neutral pronouns (Pinheiro, 2020) (sometimes referred to as *neo-pronomes* in Portuguese). These neopronouns are preferred by many individuals to refer to themselves (Miranda, 2020), but can also be used to refer to a mixed-gender group of people avoiding the default masculine plural – for example, *um grupo de alunes* instead of *um grupo de alunos* (**EN:** *a group of students*).

Although the lack of gender inclusion in NLP datasets is a multilingual problem (Zhou et al., 2019), most efforts being made towards the processing of gender-neutral language are focused on Germanic languages (Vanmassenhove et al., 2021; Sun et al., 2021; Amrhein et al., 2023). As such, we set out to create NLP models that allow for the processing of Portuguese gender-neutral language in the same way that is currently being achieved for other languages. The three main contributions of our work consist of:

- As far as we concern, the first Portuguese parallel datasets explicitly containing gender-neutral language and neopronouns, made publicly available[1], as well as a manually curated test set of 500 sentences. These datasets are comprised of sentences belonging to five different text categories: literary texts, journalistic texts, dialogues, social media posts and comments, and simpler sentences. This allows for the evaluation of Portuguese gender-neutral rewriters in different contexts. We hope this contribution will increase visibility of gender-neutral language and neopronouns in the landscape of Portuguese NLP datasets, as well as allow for future research regarding these topics;

- A rule-based gender-neutral rewriter based on

---

[1] https://github.com/leonorv/pt-gn-datasets

handcrafted rules, for which we provide open access[2];

- A neural gender-neutral rewriter[3] developed via fine-tuning a large multilingual machine translation model. This method requires relatively smaller sized datasets (when compared to training a model from scratch), and thus allows for the development of gender-rewriters for lower-resource languages.

## 2 Related Work

As previously mentioned, our system falls into the category of what we will call *gender-neutral rewriter* systems. These rewriters take as input some form of gendered text and output a gender-neutral version of that same text. What is considered to be *gender-neutral* greatly varies between proposed systems and languages.

### 2.1 Gender-fair rewriting for English

Gender-neutral rewriting is currently modeled as a text generation problem, solved with sequence-to-sequence rule-based or neural models trained with parallel data.

Vanmassenhove et al. (2021) and Sun et al. (2021) have developed similar rewriters in terms of approach and model architecture. They propose two models: a rule-based rewriter, and a neural model. The neural model is trained on a parallel dataset composed of the original, binary-gendered version of the dataset, and the respective output of the rule-based rewriter. The neural model developed by Vanmassenhove et al. (2021) outperforms their rule-based rewriter, while the rule-based rewriter developed by Sun et al. (2021) outperforms their neural model. Both models from all authors achieve a WER (Woodard and Nelson, 1982; Morris et al., 2004a) of less than 1% in the respective test sets.

In these approaches, a gender-fair version of a binary-gendered dataset is synthetically generated through a rule-based pipeline. This method, which we employ in our work, can be referred to as *forward augmentation* (Amrhein et al., 2023). Similarly to Vanmassenhove et al. (2021) and Sun et al. (2021), we train our neural model on data generated by the rule-based model. However, while the cited authors train a neural model from scratch,

we rely on fine-tuning a large pre-trained model, described in section 5.2.

### 2.2 Gender-fair rewriting for Other Languages

Alhafni et al. (2022) developed a gender rewriter for Arabic, combining both rule-based and neural models. The authors used the Arabic Parallel Gender Corpus (APGC), which contains gender labels for each word specifying whether the word is gendered or not, female or male, and referring to the first-person or the second-person. The target gender label is appended to the input words as a special token and the assumption is that the neural model pays attention to the label to output the correct gender alternatives. Section 2.3 details other gender-identifying systems that rely upon gender-tagged data.

Amrhein et al. (2023) propose a novel approach to the gender-rewriting task. While the previous models rely on forward augmentation, this approach relies on *backward augmentation* and round-trip translation to create a parallel dataset. This approach is used to create an English rewriter that matches or outperforms the results of Vanmassenhove et al. (2021). The backward augmentation approach consists of retrieving gender-fair data from large monolingual corpora and creating a rule-based pipeline to derive artificially biased text. The round-trip translation approach relies on the fact that most current machine translation models are socially biased. This can be exploited by using a biased model to translate from gender-fair text to a pivot language. This output is then translated back to the original language, creating a biased version of the original gender-fair text. The authors use the round-trip translation method to create a German rewriter, using English as a pivot language.

We identify two issues with the round-trip translation approach when developing a rewriter for the Portuguese language:

- The lack of very large Portuguese monolingual datasets containing gender-fair language;

- The lack of consistency of the existing gender-fair data regarding the usage and choice of neopronouns: due to the diversity of Portuguese gender-neutral language proposals and neopronouns (detailed in section 3), it is often the case that real examples of gender-neutral language are not consistent in terms

---

[2]https://github.com/leonorv/pt-gender-neutralizer
[3]https://huggingface.co/leonorv/pt-neural-gender-neutralizer

of gender agreement. Examples of this phenomenon are depicted in Table 6, in Appendix A.1.

Amrhein et al. (2023) resort to LLMs in order to generate additional gender-fair examples. We address this approach as future work in section 7.

## 2.3 Other Gender-Identifying Systems

At the time of writing, no gender-neutral rewriters have been developed for Romance languages. However, Jain et al. (2021) have developed a neural machine translation system, for the Spanish language, trained to translate from one (binary) gender alternative to another, as a method for generating augmented data. This task is very similar to gender-neutral rewriting, since it requires identifying and rewriting gendered terms. The parallel dataset is enriched with *re-genderablility tags*, i.e., if the source and the target parts differ, then the sentence is considered *re-genderable*. Using these tags results in a reduced error rate when compared to a standard neural system (without any tags).

Bellandi and Siccardi (2022) have developed a system that identifies gender-discriminatory language for Italian. The authors describe two possible ways in which language can be gender discriminatory: when sentences contain "only the male form of a noun having a different female form", and "sentences containing nouns having the same male and female form, without any other grammatical element to stress reference to both genders". The latter case is also common in Portuguese (e.g. using *os docentes* with the male article *os*, when *docentes* is both the male and female term for "teacher"). The authors trained a neural model to recognize these situations, assigning a label to each.

As far as we are aware, datasets that contain gender tags are not available for Portuguese, and therefore an approach that relies on gender tags (such as the approaches of Bellandi and Siccardi (2022), Jain et al. (2021), and Alhafni et al. (2022)) is currently not viable.

## 3 Grammatical Gender and Gender-neutral Language in Portuguese

Like most Romance languages, Portuguese has two genders: masculine (characterized by an -o termination in gendered terms) and feminine (characterized by an -a termination in gendered terms). Most of the current proposals for a third neural gender for Portuguese are characterized by an -e or an -u termination in gendered terms. These proposals tend to arise in the form of practical guides, both by Brazilian and European Portuguese authors, and tend to be based on informal studies and observations of the neutral language used by queer communities. Since the usage of gender-neutral language in Portuguese is an ongoing debate, in this section we present some guides reporting on the community usage of gender-neutral language.

### 3.1 Proposals for a Neutral Grammatical Gender Within Portuguese

The earliest source we were able to find is the "Manifesto ILE" (P. Berlucci, 2015), which suggests the gender-neutral *ile/dile* pronouns. However, the authors of this guide do not provide any rules or recommendations on the usage of this pronoun and agreement with nouns and adjectives.

Caê (2020) presents the *Elu*, *Ile*, *Ilu*, and *El* neopronoun systems. While these systems differ in terms of the pronoun in use, the agreement with nouns and adjectives is similar, requiring an -e termination. For example, *filho* (**EN:** *son*) becomes *filhe*. The author also provides some general tips for rephrasing sentences to omit gender. For example, *ela caiu* (**EN:** *she fell*) becomes *aquela pessoa caiu* (**EN:** *that person fell*).

Marques and Santos (2021) recommend the usage of the neopronoun *éle/déle* and the termination system -e. An example sentence formed with the rules from this guide is: *O professor deu as boas-vindas a todes es alunes.* (**EN:** *The teacher welcomed all the students.*). The authors also recommend the usage of already existent neutral forms – e.g., *monarca* (**EN:** *monarch*) instead of *rei/rainha* (**EN:** *king/queen*).

The only other European Portuguese guide we found sets rules for a system using the pronoun *elu/delu* (Valente, 2020). The authors provide general grammatical rules and example sentences, such as *Aquelu menine é minhe filhe* (**EN:** *That kid is my child*).

Most of the systems we observed converge on how the agreement of the neutral pronoun with nouns and adjectives should be done. For example, the neutral termination tends to be -e (e.g. "filho" → "filhe"), since -e is a vowel that provides contrast in speech with -o and -a. An exception is often made for terms whose masculine form ends with -e: in those cases, the corresponding neutral

form ends with -u in order to be distinguished from the masculine form (e.g. "aquel**e**" → "aquel**u**"). An example sentence using different neopronouns can be observed in Table 8.

## 3.2 Rejection of -x and -@ Terminations

The characters -x and -@ have also been adopted as gender-neutrality markers (eg: *todxs* or *tod@s*), but there is concern with the legibility and usability of said terminations. Due to people and reading softwares not being able to pronounce words that terminate in -x or -@, their usage has been categorized as ableist (P. Berlucci, 2015; Marques and Santos, 2021; Valente, 2020). For these reasons, these terminations tend to be rejected by the community. Bearing this in mind, in this work we focus on the implementation of rules based on -e and -u terminations.

## 4 Datasets

Our dataset is split into five text categories:

**Literary Texts**   Selected works found in the *DIP* collection[4] from Linguateca[5] resource center. DIP is a shared task whose goal is to identify of characters and respective attributes in literary works (Santos et al., 2022). In order to avoid examples with an orthography that might be too different from modern Portuguese, we have only selected works released after 1910. Cleaning and processing the data consisted of the removal of tags (such as chapter indications, language tags, etc.) and author notes.

**Journalistic texts**   Random sample of sentences found in the *NaturaPublico94* dataset, from Projecto Natura[6]. The original corpus contains the first 2 paragraphs of each article in the Portuguese newspaper *Público*, retrieved during the period of 1991 to 1994. *NaturaPublico94* was retrieved during 1994.

**Dialogues**   Random sample of extracted sentences from the *SubTle* corpus (Ameixa and Coheur, 2013; Ameixa et al., 2014). *SubTle* aggregates dialogues from movie subtitles, extracted from IMDB[7] and pertaining to one of four movie genres: Horror, Scifi, Western, and Romance. During cleaning of the dataset, we removed speaker tags.

**Social Media**   Random sample of tweets from the *Portuguese Tweets for Sentiment Analysis*[8] dataset, which contains examples retrieved mainly from 01/08/2018 to 20/10/2018. We only used tweets from the "no theme" partition of the dataset. We removed links, mentions, hashtags, and emojis.

**Simple Sentences**   Samples from the Portuguese dataset of the Tatoeba (Tiedemann, 2020) corpus, consisting of relatively simpler (both in terms of vocabulary and syntactic construction) and less noisy sentences.

We chose to divide the data into five categories to analyze the performance of our models in different types of text, and how they account (or not) for variability of sentence structure and vocabulary.

## 4.1 Automatically Curated Set

We curated sets of 5,000 sentences from each text category, amounting to a total of 25,000 examples. The gender-neutral alternatives of each example are generated by the rule-based model. This parallel dataset, containing the original (binary-gendered) sentences and the respective gender-neutral version, was used for training our neural model. Table 9 (from Appendix A.4) depicts one example for each dataset category. This automatically curated set is composed of approximately 60% gendered (14874 sentences) and 40% non-gendered sentences (10126 sentences). Due to the novelty of this line of research, the impact of the percentage of gendered sentences on the performance of gender-rewriter models is still understudied. We leave that particular topic for future work.

## 4.2 Manually Curated Test Set

For curating our test set, we selected an additional 100 sentences from each of the five categories described in section 4. All 500 examples are *gendered*, containing either named entities, human referents, or personal pronouns.

The 500 sentences in the collection were manually annotated by an NLP researcher. A sample of 100 examples belonging to the collection was annotated by other 5 fellow researchers in order to calculate the annotator agreement. The annotators followed an annotation guide for the *elu* system,

---

[4]https://www.linguateca.pt/aval_conjunta/dip/colecao.html
[5]https://www.linguateca.pt/
[6]https://natura.di.uminho.pt/ jj/pln/corpora/
[7]https://www.imdb.com/

[8]https://www.kaggle.com/datasets/augustop/portuguese-tweets-for-sentiment-analysis

whose rules are consistent with the ones employed in our rule-based model and are inspired by the proposals presented in section 3.1, particularly the ones authored by Caê (2020) and Marques and Santos (2021). A sample of the said guide can be found in Appendix C.

We calculate the agreement using the metrics WER, CER, and exact match. We achieve a WER of 2.15%, a CER (Morris et al., 2004b) (character error rate) of 0.54%, and an exact match score of 82%. We assume that these results reflect the quality of the full collection.

Annotation disagreements often arise either from the existence of several possible gender-neutral alternatives, or from uncertainty over if a certain term should be neutralized. Examples of these types of disagreements are found in Table 1. In the first sentence, the devil (*diabo*) may be considered a genderless entity, and therefore terms related to it should not be rewritten. However, the female form of *diabo*, *diaba*, can be used, which may be used as an argument in favor of *diabo* being a gendered term. In sentence 2, the genderless term *doentes* is a synonym to the gender-neutral term *enfermes*, derived from the term *enfermos/enfermas*.)

| Annotator X | Annotator Y |
|---|---|
| Está **o diabo** à solta.[9] | Está **ê diabe** à solta. |
| Ês **enfermes** eram, muitas vezes, expostes na rua.[10] | Ês **doentes** eram, muitas vezes, expostes na rua. |

Table 1: Annotator disagreements are marked in bold.

# 5 Models

## 5.1 Rule-based Model

As depicted in Figure 3 (Appendix B.1), the rule-based model (RBM) is composed of three main modules, described in the following sections.

### 5.1.1 Preprocessing Pipeline

The preprocessing pipeline consists of tokenization, POS-tagging, dependency parsing, and named entity recognition models made available by the Stanza (Qi et al., 2020) toolkit. At the time of writing, Stanza does not make available any Portuguese named entity recognition model. Therefore, the preprocessing pipeline currently makes use of the Stanza named entity recognition model for Spanish, chosen due to the similarity between the two languages.

### 5.1.2 Human Referents Extractor

The extractor module sweeps the input text to find all proper nouns, nouns, and pronouns that refer to people. Their positions in the text, as well as the positions of their heads in the dependency parsing graph are then stored and used in the rewriter module. This task requires access to a Portuguese wordnet (Miller, 1995). We have chosen OpenWordnet-PT (de Paiva et al., 2012), due to its integration with the NLTK library (Loper and Bird, 2002). NLTK allows access to the lexicographer file of each word sense. Lexicographer files are split into 45 categories. Category 18 contains "nouns denoting people". Our extractor module checks if the lexicographer file name of the current synset corresponds to this category. If so, that noun is referent to a human, and, as such, terms related to it should have the correct gender form.

### 5.1.3 Rewriter

The final module of the RBM rewrites gendered terms related to human referents, whose positions in the sentence are stored by the extractor module.

In Portuguese, nouns, pronouns, clitic pronouns, determiners, adjectives, and verbs can be gendered. The rules used for "gender-neutralizing" each term depend on their word class, as described in further detail below.

**Nouns & Pronouns** Since the neopronoun "elu" is the most used in the third neutral gender proposal we have presented in section 3, the system we employ in our model currently only uses that specific neopronoun. Certain gendered nouns have an existent gender-neutral synonym, which can be used instead of rewriting the gendered noun using rules. The rewriter module performs a lookup to a table (depicted in Appendix A.2) containing some of these gender-neutral terms and uses them accordingly.

**Clitic Pronouns** If a clitic pronoun is gendered, then it is rewritten (e.g. "Eu vou vê-**la**."[11] → "Eu vou vê-**le**.".

**Determiners** Definite articles that precede a proper noun are omitted (e.g. "**O** João é fe-

---

[9]**EN:** "The devil is on the loose."
[10]**EN:** "the sick were often exposed in the street."

[11]**EN:** "I'm going to see her."

liz."[12] → "João é feliz."). Other types of determiners are neutralized if their head in the dependency graph is referent to a human (e.g "João é **um** rapaz."[13] → "João é **ume** jovem.").

**Adjectives**   The task of checking whether an adjective refers to a certain referent is complex, since an adjective can be in multiple positions in a sentence. The rule-based rewriter module assumes that an adjective should be rewritten if either the adjective itself or its head in the dependency graph has been marked by the extractor module as either referent to a human or a head of a term marked as referent to a human. This rule tends to correctly rewrite adjectives in sentences with a relatively simple construction. However, it fails in sentences where an adjective and a human-referent noun share a root term in the dependency graph. This is illustrated in Figure 1 and Figure 2.

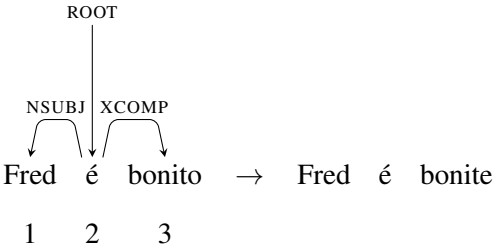

Figure 1: Since both the terms *Fred* and *bonito* share a head node, *é*, and *Fred* is a human referent, the adjective is rewritten. This sentence can be translated as "Fred is pretty."

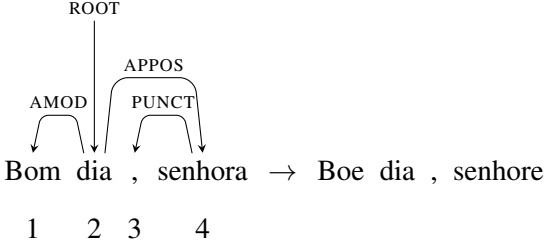

Figure 2: The same rule that worked for the case on Figure 1 fails with this sentence structure. Since *senhora* is a human referent, the adjective *bom* is incorrectly neutralized, even though it refers to the term *dia*. This sentence can be translated as "Good morning, madam!"

**Verbs**   Most verb tenses are not gendered, with the exception of some main verbs that require an auxiliary verb, such as past participle forms (e.g

---

[12]**EN:** "João is happy."
[13]**EN:** "João is a boy."

"João foi levad**o** a jantar."[14]  → "João foi levad**e** a jantar."). The RBM rewrites verb forms that require an auxiliary verb different from the verb "ter" (to have), since that particular verb is used in several non-gendered tenses. We make an exception for gerund, which can require an auxiliary verb different from "ter", but is not a gendered form (e.g "João foi andando."[15] → "João foi andando.").

## 5.2   Neural Model

We fine-tuned the M2M100 (Fan et al., 2021) multilingual encoder-decoder, setting both the source and target languages as Portuguese. For training, we used the original sentences of our automatically curated set as source, and the respective versions rewritten by the RBM as target. We followed an 80-10-10 split for training, validation, and test sets. For more details about the chosen hyperparameters and model please refer to Appendix B.

## 6   Experiments

### 6.1   Metrics

Due to the novelty of the task of gender-neutral rewriting, we find that there is a lack of general consensus on useful metrics for evaluating these types of rewriter systems.

For comparison with the works described in section 2, we employ WER and BLEU-4 (Papineni et al., 2002) (usually the default n-gram value for BLEU scores) as metrics. Since gender-neutral language in Romance languages often consists in a single character change (e.g. "João é bonit**o**."[16] → "João é bonit**e**."), we also employ CER (character error rate) as a metric. Additionally, we calculate BLEU-1 and ROUGE-1 (Lin, 2004).

### 6.2   Results

We tested the RBM and the NM on our manually curated test set. We compare the results against a baseline metric (Base), which computes the metrics between the original examples and the gender-neutral versions.

The results are detailed in Table 2. The superior performance of the neural model on the dataset with simpler sentences suggests that the neural approach may be better suited for rewriting source sentences with a simpler syntax. While the average performance of the rule-based model tends to

---

[14]**EN:** "João was taken out to dinner."
[15]**EN:** "João walked."
[16]**EN:** "João is pretty."

be slightly better compared to the performance of the neural model, the higher ROUGE score of the neural model suggests that this approach tends to rewrite fewer terms, since a higher ROUGE score corresponds to a higher rate of true positives.

### 6.3 Discussion

**Rule-based Approach**   If the RBM produces a wrong output, the error can arise from one and/or two circumstances:

- **One of the preprocessing pipelines has produced an error:** for instance, the named entity recognition model may fail and wrongly tag a noun as a human referent, as in the case of the first example found in Table 4. Since we are using a Spanish NER model, this is a common mistake with names that may be unusual in Romance languages (such as "Bill", found in the fourth example found in Table 4.) It might be worth to note that this problem is not specifically due to the NER model being focused on the Spanish language, but due to it being focused on a Romance language. These types of errors are not directly caused by the rules we have defined, and as such can only be mitigated by using different preprocessing tools.

- **The rewriter rules have failed:** The rules we have found to be most susceptible to errors are the ones regarding *adjectives*, as we have described in section 5.1.3. Since Portuguese adjectives are diverse in terms of gendered terminations, creating rules that encompass all adjectives in the language may be complex and time-consuming.

**Neural Approach**   If the NM produces a wrong output, the error can arise from:

- **The model has learned a wrongly rewritten form of a certain term:** as is the case in the incorrect sentence depicted in Table 5. These types of errors may be mitigated by improving the quality of the rewriting rules.

- **The model is either ignoring terms that should be rewritten, or rewriting terms that should be ignored:** we have noticed that adjectives are more susceptible to these types of errors, most likely due to the model simply learning how to rewrite a certain term, but not

capturing if the term is related (or not) to a human referent.

Similarly to Vanmassenhove et al. (2021), we have found that our neural model is able to generalize over the training data. For instance, the fifth sentence of Table 4 contains the expression *soprador de apito* (**EN:** *whistle blower*). Although the word *soprador* does not exist in the model training data, the model is able to rewrite the term as the respective gender-neutral form *sopradore*. However, this is not always the case: in the first sentence of Table 3, the neural model is not able to correctly rewrite the term *ricas* (**EN:** *rich*). We hypothesize that this may be either due to the higher complexity of the respective gender neutral form (*ricas → riques*) or due to the lack of representation of female gender forms in data (both in our own datasets and the training data of our base neural model).

## 7  Conclusion

In this paper, we present the first Portuguese dataset explicitly containing gender-neutral language, along with a rule-based and a neural gender-neutral rewriters. Additionally, we provide a manually annotated collection of 500 original sentences and a respective gender-neutral version. One entry of our automatically curated dataset (used for training the NM) consists of a binary-gendered sentence with the respective gender-neutral version provided by the RBM. We provide the first benchmarks of the gender-neutral rewriting task for the Portuguese language.

Although the neural model can generalize over the seen data, we hypothesize that it fails to internalize the context of sentences and whether gendered terms refer to humans or objects.

In future work, we hope to bypass this issue either by training a larger model with more quality data, and/or developing a hybrid rule-based/neural model. Another possible approach is data augmentation using LLM prompting. In a first attempt to replicate this approach, we have prompted ChatGPT[17], the sibling model to InstructGPT (Ouyang et al., 2022), to generate Portuguese gender-neutral sentences. Results are depicted in Appendix D. Although a promising approach, we hypothesize that the generated sentences do not have the necessary consistency and variety to create a quality dataset. We expect that, in the future, with the advancement

---

[17]chat.openai.com

|  |  | Literary | Journalistic | Dialogue | Social Media | Simple | Average |
|---|---|---|---|---|---|---|---|
| **WER%** | Base | 15.07 | 15.43 | 21.93 | 14.22 | 26.45 | 18.63 |
|  | RBM | 7.89 | **4.98** | **7.95** | **5.27** | 7.02 | **6.62** |
|  | NM | **7.56** | 8.03 | 8.19 | 7.35 | **6.2** | 7.47 |
| **CER%** | Base | 3.90 | 2.86 | 7.20 | 3.75 | 7.36 | 5.01 |
|  | RBM | 1.63 | **1.03** | **2.26** | **1.23** | 1.72 | **1.57** |
|  | NM | **1.55** | 2.61 | 2.47 | 1.72 | **1.39** | 1.95 |
| **BLEU-4** | Base | 57.90 | 71.79 | 42.93 | 65.97 | 32.89 | 54.30 |
|  | RBM | 76.28 | **87.36** | **72.92** | **80.40** | 74.37 | **78.27** |
|  | NM | **79.77** | 83.91 | 71.60 | 78.53 | **76.72** | 78.106 |
| **BLEU-1** | Base | 81.00 | 85.13 | 74.53 | 83.72 | 71.23 | 79.12 |
|  | RBM | 90.75 | **94.32** | **90.75** | **93.25** | 93.00 | **92.414** |
|  | NM | **92.11** | 92.65 | 90.70 | 91.74 | **93.83** | 92.21 |
| **ROUGE-1** | Base | 83.15 | 90.80 | 79.15 | 86.63 | 74.41 | 82.93 |
|  | RBM | 91.67 | **95.00** | 91.04 | **93.77** | 93.38 | 93.17 |
|  | NM | **92.95** | 93.78 | **92.35** | 93.38 | **94.59** | **93.41** |

Table 2: Metrics for the manually curated test set for each data category. The best model for each category/metric pair is marked in bold.

| RBM | NM |
|---|---|
| Vocês são **riques**. | Vocês são **ricas**. |
| Elu está **parade** na parte mais **fria** do complexo. | Elu está **parado** na parte mais **frie** do complexo. |
| [...] foram convidadas **dues** profissionais **vindes** da RTP [...]. | [...] foram convidadas **duas** profissionais **vindas** da RTP [...]. |
| Em 1989 conseguiram eleger deputades para **o Parlamento Europeu**. | Em 1989 conseguiram eleger deputades para **Parlamento Europeu**. |
| Oi sou a sub **delegade** | Oi sou a sub **delegada** |
| Caso **Haddad** seja eleito, eu vou fazer diferente de alguns e vou pensar positivo [...] | Caso **o Haddad** seja eleito, eu vou fazer diferente de alguns e vou pensar positivo [...] |

Table 3: Example sentences where the rule-based model performs better than the neural model.

| RBM | NM |
|---|---|
| Irene é **de** Peru. | Irene é **do** Peru. |
| Elu é a **maiora** pessoa que já viveu. | Elu é a **maior** pessoa que já viveu. |
| Eu dormi com a vítima número **dues**. | Eu dormi com a vítima número **dois**. |
| Minhe parente não me deixa sair com **o Bill**. | Minhe parente não me deixa sair com **Bill**. |
| [...] Tom trabalhava como **soprador** de apito [...]. | [...] Tom trabalhava como **sopradore** de apito [...]. |
| **Boe** dia criança !!! | **Bom** dia criança !!! |

Table 4: Example sentences where the neural model performs better than the rule-based model.

|  | RBM | NM |
|---|---|---|
| Correct | **Ê presidente** se reunirá amanhã com **ês empresáries** mais importantes do país. | **Ê presidente** se reunirá amanhã com **ês empresáries** mais importantes do país. |
| Incorrect | Você é muito **gentile**. | Você é muito **gentile**. |

Table 5: Example sentences where both models produce the same output (correctly or incorrectly).

of these types of models and optimized prompting, we are able to generate quality gender-neutral examples that allow us to create larger inclusive datasets.

## Limitations

The usage of gender-neutral language in Portuguese-speaking communities is a diverse and ever-changing linguistic phenomenon. While we present some of the third neutral gender pronouns found in literature in section 3, our models only process the neopronoun *elu* and follow rewriting rules that are not universally agreed upon.

Another limitation is that our rule-based model suffers from low scalability to long text. Handmade rules often fail to correctly rewrite long sentences due to their more complex and unpredictable structure.

While using pre-trained large multilingual translation models may be an option for developing gender-neutral rewriters for lower-resource languages, this method is dependent on the existence of said models for the target language. Languages with very low resources are often not represented in such models.

## Ethics Statement

Our publicly available datasets were not filtered for harmful or hateful content. While we fine-tuned our neural model on this data, since the output of our neural model consists of the input text with minor changes, we do not believe it is susceptible to hallucinations of harmful text.

The annotation was voluntary and the participants received no financial compensation for the task.

We do not claim that any Portuguese gender-neutral grammar system, gender-neutral termination, or neopronoun is in any way superior or should be used to detriment of others. In future work, we expect to develop models that accommodate the addition of new systems.

## Acknowledgements

We would like to thank the voluntary annotators for their contributions.

This research was supported by the Portuguese Recovery and Resilience Plan through the project C645008882-00000055 (Center for Responsible AI), and through *Fundação para a Ciência e a Tecnologia* (FCT), specifically through the INESC-ID multi-annual funding with reference UIDB/50021/2020.

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

## A Gender-neutral Grammar Details and Examples

### A.1 Examples from Users of Portuguese Gender-neutral Language

| |
|---|
| Algum amigue pelo litoral de cabedelo para um rolê na praia? |
| Minha amigue tirou a runa das bruxas para mim e eu estou impactada |
| Todas e todes recebendo tratamento igualitário. . . |
| Procuro um namorade que me dê carinho e atenção |

Table 6: Example excerpts retrieved from Twitter in 20/02/2023. We slightly modified the examples to lower searchability and increase the privacy of the authors.

### A.2 Gender-neutral Expressions

| Original Expression | Gender-neutral Expression |
|---|---|
| homem/mulher | pessoas |
| rapaz/rapariga | jovem |
| menino/menina | criança |
| pai/mãe | parente |
| aluno/aluna | estudante |
| professor/professora | docente |
| esposo/esposa | cônjuge |
| rei/rainha | monarca |

Table 7: Binary-gendered expressions and respective gender-neutral alternative expressions.

### A.3 Usage of Different Portuguese Neopronouns

| Personal Pronoun | "Ele comeu a pizza dele.[18]" |
|---|---|
| elu | "Elu comeu a pizza delu." |
| ile | "Ile comeu a pizza dile." |
| ilu | "Ilu comeu a pizza dilu." |
| éle | "Éle comeu a pizza déle." |
| el | "El comeu a pizza del." |

Table 8: Usage of different Portuguese neopronouns.

---

[18]**EN:** "He ate his pizza."

### A.4 Dataset Examples

Table 9 depicts examples for each dataset category.

## B Model Details

### B.1 Rule-based Model Pipeline

Figure 3 depicts the three main modules and respective dependencies of our RBM.

### B.2 Hyperparameter Search Results

We performed automated hyperparameter search using Weights & Biases (Biewald, 2020). We ran a total of ten sweeps, exploring different combinations of values for learning rate and weight decay. The results are depicted in Figure 4.

### B.3 Training Hyperparameters

We fine-tuned the M2M100_418M model[19] for 5 epochs, using a learning rate of 0.00005569, a weight decay of 0.02, batch size of 8, and eval batch size of 8. We use a maximum target length of 128. The rest of the parameters are left as Hugging-Face Seq2SeqTrainingArguments[20] defaults. We used the Seq2SeqTrainer[21] class to complete the fine-tuning procedure, using *sacrebleu* (Post, 2018) for our the compute_metrics fuction.

## C Annotation Guidelines Sample

The annotation guidelines contain rewriting rules for every gendered word class. For the sake of brevity, in this section we present only the rules for determiners (*determinantes*, in Portuguese) and contractions with prepositions (*contrações com preposições*). Although the original guide was written in Portuguese, here we also present an English version of the text.

### C.1 Determinantes

**PT** Se precedidos por um nome próprio, os determinantes artigos definidos (o, a, os, as) devem ser omitidos. Se precedidos por um nome comum, não devem ser omitidos. Os determinantes artigos indefinidos (um, uma, uns, umas) nunca são omitidos e, se referentes a uma pessoa, devem ser reescritos com a respetiva forma neutra.

---

[19]https://huggingface.co/facebook/m2m100_418M
[20]https://huggingface.co/docs/transformers/main_classes/trainer#transformers.Seq2SeqTrainingArguments
[21]https://huggingface.co/docs/transformers/main_classes/trainer#transformers.Seq2SeqTrainer

| Category | Original Sentence | Gender-neutral Sentence |
|---|---|---|
| Literary | É orgulhoso e de opinião, como ele só! (**EN:** *He is proud and opinionated, as only he can be!*) | É orgulhose e de opinião, como elu só! |
| Journalistic | Fomos à procura deles e organizámos um almoço comemorativo. (**EN:** *We went looking for them and organized a celebratory lunch.*) | Fomos à procura delus e organizámos um almoço comemorativo. |
| Dialogue | Precisa saber só de olhar para a mulher, sem ela dizer. (**EN:** *He needs to know just by looking at the woman, without her saying so.*) | Precisa saber só de olhar para a pessoa, sem elu dizer. |
| Social Media | foi ela quem fez o exorcismo. (**EN:** *she was the one who performed the exorcism.*) | foi elu quem fez o exorcismo. |
| Simple Sentences | Eu estou viciado em mascar chiclete. (**EN:** *I'm addicted to chewing gum.*) | Eu estou viciade em mascar chiclete. |

Table 9: Dataset categories and respective examples. The original sentences contain idiomatic expressions, which we tried to capture in the English translation.

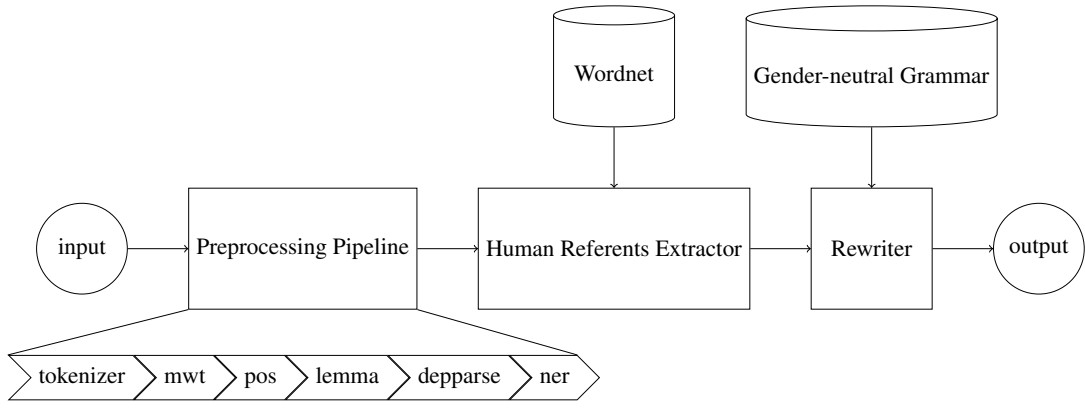

Figure 3: RBM Pipeline.

**EN** If preceded by a proper noun, the definite article determiners (o, a, os, as) must be omitted. If preceded by a common noun, they must not be omitted. The indefinite article determiners (um, uma, uns, umas) are never omitted and, if they refer to a person, must be rewritten according to the respective neutral form.

**o(s)/a(s) → ê(s)/omitido**

- O Sérgio é o amigo da Mariana.[22] → Sérgio é ê amigue de Mariana.

- O João é simpático.[23] → João é simpátique.

**um/uma(s) → ume(s)**

---
[22]**EN:** "Sérgio is Mariana's friend."
[23]**EN:** "João is nice."

- O Miguel é um escritor.[24] → Miguel é ume escritore.

### C.2 Contrações com Preposições

**PT** Os casos específicos das contrações **pelo/pela** e **ao/à** possuem uma particularidade: no caso de serem precedidas por um nome próprio, deve-se optar pela 1ª forma neutra apresentada (**por** ou **a**). No caso de serem precedidas por outro tipo de nome, deve-se optar pela 2ª forma neutra apresentada (**pele** ou **ae**).

**EN** For the particular cases of the contractions **pelo/pela** and **ao/à**: in case they are preceded by a proper noun, one should opt for the 1st presented neutral form (**por** or **a**). If they are preceded by

---
[24]**EN:** "Miguel is a writer."

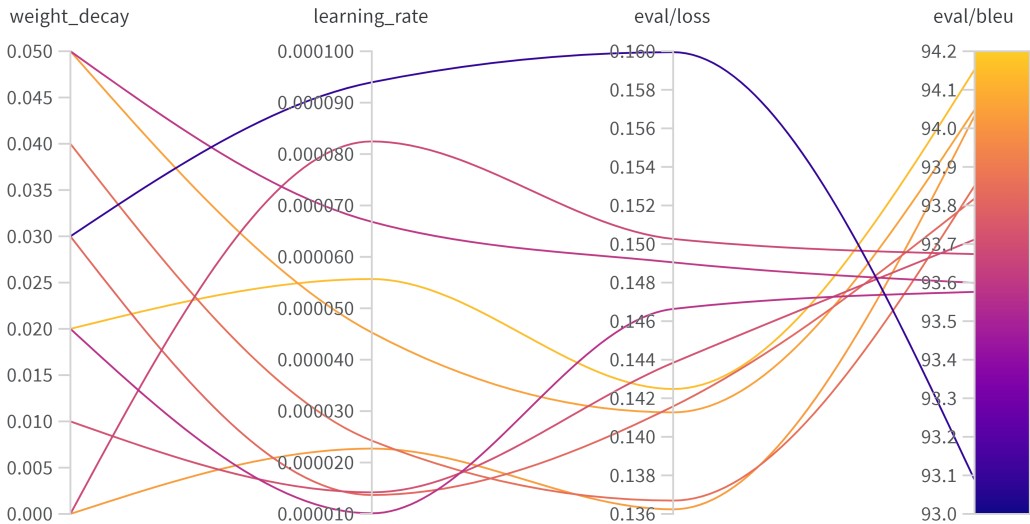

Figure 4: Hyperparameter search results, provided by Weights & Biases (Biewald, 2020). The best configuration consists of a weight decay of 0.02 and a learning rate of 0.00005569.

another type of noun, one must choose the 2nd neutral form presented (**pele** or **ae**).

**pelo(s)/pela(s) → por/pele(s)**

- A Mariana fez tudo pela Lúcia.[25] → Mariana fez tudo por Lúcia.

- A Mariana fez tudo pelas amigas.[26] → Mariana fez tudo peles amigues.

**ao(s)/à(s) → a(s)/ae(s)**

- A Teresa deu o livro ao Pedro.[27] → Teresa deu o livro a Pedro.

- A Teresa deu o livro ao amigo.[28] → Teresa deu o livro ae amigue.

## D  LLM Prompting

**Prompt**  "Generate 5 examples of Portuguese gender-neutral sentences, following the "elu" system. Bear in mind that, in the "elu" system, terms that refer to humans tend to have an -e termination. Make sure to use sentences that include nouns, adjectives, and verbs."

**Generated Sentences**  Wrong gender-neutral terms or terms whose gender agreement is not correct are marked in bold.

- Elu é **um** estudante dedicade. (**EN:** *They are a dedicated student.*)

- Elu adora explorar novos lugares e culturas. (**EN:** *They love exploring new places and cultures.*)

- Elu trabalha como **profissionale** da área de tecnologia. (**EN:** *They work as a technology professional.*)

- Elu é uma pessoa **criative** e **talentose**. (**EN:** *They are a creative and talented person.*)

- Elu está **escrevende** um livro incrível. (**EN:** *They are writing an amazing book.*)

---

[25]**EN:** "Mariana did everything for Lúcia."
[26]**EN:** "Mariana did everything for her friends."
[27]**EN:** "Teresa gave Pedro the book."
[28]**EN:** "Teresa gave the book to her friend."