# OpenReview forum: "A Rewriting Approach for Gender Inclusivity in Portuguese"
_EMNLP/2023/Conference — EMNLP 2023 Findings_

### Official Review · Reviewer_KYF6 · 2023-07-24

**Soundness:** 4

**Excitement:**

3: Ambivalent: It has merits (e.g., it reports state-of-the-art results, the idea is nice), but there are key weaknesses (e.g., it describes incremental work), and it can significantly benefit from another round of revision. However, I won't object to accepting it if my co-reviewers champion it.

**Paper Topic And Main Contributions:**

This paper is about gender-neutral rewriting for the Portuguese language. The authors describe the specific difficulties that Portuguese poses for gender-neutral languages. They then apply some established techniques, namely a rule-based rewriter and a paraphrasing model trained on augmented data, to a new, manually created test set across five domains. An automatic and a qualitative evaluation indicate that the neural approach shows some generalization, but is overall less accurate than the rule-based approach.

**Reasons To Accept:**

- Valuable resources: The papers provides new resources for gender-neutral rewriting in Portuguese: a small-scale, manually created test set as well as two different rewriting systems (I could not verify the resources because the code and data were not provided to me).
- Linguistic expertise: The paper presents information on grammatical gender and gender-neutral language in Portuguese, which is helpful for future work.
- Clear presentation: The paper is well-written and easy to follow.


**Reasons To Reject:**

- Limited technical novelty: The main contribution of the paper are resources and baseline experiments for gender-neutral rewriting in Portuguese, for which such resources did not previously exist. The paper does not make novel technical (e.g., algorithmic or modeling) contributions (nor does it claim to do so).

**Reproducibility:**

5: Could easily reproduce the results.

**Reviewer Confidence:**

4: Quite sure. I tried to check the important points carefully. It's unlikely, though conceivable, that I missed something that should affect my ratings.

---

> ### Author Rebuttal · Authors · 2023-08-28
>
> Thank you for your feedback. If the paper is accepted, we will take into account all your comments.
>
> You are right, we do not claim to make technical contributions. We are providing resources for the Portuguese language in the form of datasets containing gender-neutral language and gender-neutral rewriter models.

---

### Official Review · Reviewer_CAWR · 2023-08-03

**Typos Grammar Style And Presentation Improvements:** 1) I find the interchangeable usage o…
**Soundness:** 2

**Excitement:**

2: Mediocre: This paper makes marginal contributions (vs non-contemporaneous work), so I would rather not see it in the conference.

**Paper Topic And Main Contributions:**

In this paper, the authors present the first rule-based and neural-based gender-neural rewriting models for Portuguese. They built the first Portuguese parallel dataset for gender-neutral rewriting across five domains: literary texts, journalistic texts, dialogues, social media posts and comments, and simpler sentences. The dataset was created automatically by using a rule-based model which the authors built to gender-neutralize gendered sentences. The dataset contains 25000 parallel sentences (5000 sentences per domain). Further, the dataset contains 500 sentences that were manually annotated and used for testing the gender-neutralizing rewriting models' performances. Moreover, the authors demonstrate the usage of the dataset by building different gender-neutral rewriting models and evaluating their performance.








**Questions For The Authors:**

See above.

**Reasons To Accept:**

The dataset will be made publicly available and would be very useful in building gender-neutral NLP models. This will reduce the biases that are embedded with the models, and which are amplified further in downstream tasks. Moreover, the dataset constitutes the first of its kind on the problem of gender-neutral rewriting in Portuguese, which is a gender-marking language and morphologically richer than English. Further, the authors do a good job of explaining grammatical gender in Portuguese and the various ways to express gender neutrality.

**Reasons To Reject:**

I found parts of the paper to be confusing and somewhat lacking across various sections:


1. Data:
    * The authors mention that all examples in the test set are gendered, what about the examples in the so-called "development" set? Are they all gendered? If so, what is the percentage of the sentences containing at least one masculine vs feminine gendered reference? If not all the sentences are gendered, how many sentences do not contain any gender references? Adding these statistics to the dataset description would make things much clearer to the readers.

    * I also found calling a dataset a "development" set is confusing in the context of this paper. Particularly because a development set (or a dev set), has the same meaning as a "validation" set. So I would avoid using the term development to describe the dataset that encompasses both training, validation, and testing.

    * Most people reading this paper do not understand Portuguese, so including English translations within the tables and highlighting the gendered words will make things easier to read.

2) Models & Experiments:

    * Rule-based system: The authors mention that Stanza doesn't have a NER system for Portuguese so they used the Spanish NER system. Why didn't the author train a Portuguese-specific NER model instead? That should be trivial given that there are Portuguese NER datasets available. The authors in fact mention that this could lead to data errors (Section 6.3) and therefore the models would not be able to be trained and evaluated properly. This makes the quality and usability of the dataset questionable.

    * Neural Model: The authors mention that split the "development" data twice and I found that to be confusing. It's unclear to me if they follow the same splits that were used to train the rule-based and the neural models.

    * The "base" system should really be named a "Do Nothing" baseline because it basically outputs the input to the output as it is.

3) Error Analysis:

    * The authors discuss the main sources of where the errors could be coming from when evaluating the models. But they do not provide any sort of quantification on each of these sources. Providing examples indicating that the rule-based system is better than the neural model (or vice-versa) is not sufficient to understand the models' behaviors quantitatively. I would advise the authors to either do a manual error analysis on a subset of examples to quantify the types of errors across different models.



**Reproducibility:**

4: Could mostly reproduce the results, but there may be some variation because of sample variance or minor variations in their interpretation of the protocol or method.

**Reviewer Confidence:**

4: Quite sure. I tried to check the important points carefully. It's unlikely, though conceivable, that I missed something that should affect my ratings.

---

> ### Author Rebuttal · Authors · 2023-08-28
>
> Thank you for your feedback. If the paper is accepted, we will take into account all your comments.
>
> Regarding your questions and reasons to reject:
>
> 1.1 The "development set" examples are not necessarily gendered. We have trained the neural model with examples that are not necessarily gendered in the hope that it would learn which terms should be rewritten and which terms should not. We have created a gendered manually curated test set due to the core of the work being gender-neutral rewriting. Using a gendered test set helps us to easily understand the performance of the models, since these models would most likely be used with gendered sentences (in a real-world scenario). Vanmassenhove et al. (2021) have also created a gendered test set.
>
> 1.2 You are absolutely right. What we call "development set" is the dataset used during the training process of the neural model.
>
> 2.1 We have not been clear about the NER problems. Using a Portuguese NER model would not solve the data errors that are created due to the named entity recognition module. The actual problem is that we are using a Romance language NER model. For instance, the example we provide in the paper mentions the name "Bill", which is not tagged as a human referent due to it not being picked up by the NER model. A Portuguese model would most likely also not recognize "Bill", since it is an English name. Since we are creating resources for the Portuguese language, we have decided to use a NER model trained on a language similar to Portuguese, even though this may cause errors when encountering a non-Spanish/Portuguese name.
>
> 2.2 We have not been clear about the "development set" splits. Our "development set" was split into training, validation, and test sets because it is required by the HuggingFace framework we have used during the neural model's training process. Those splits are different from what we call "Test Set", described in Section 4.2. Our test set consists of 500 manually curated gendered sentences, and was used for evaluating both the neural model and the rule-based model. There is no intersection between the "development set" and the test set.
> Note: the rule-based model was not trained (due to its rule-based nature).
>
> 2.3 We have named the "Do Nothing" system "Base", because it is the term used by Vanmassenhove et al. (2021). We have found that authors of gender-neutral rewriters tend to report the results of  "doing nothing" for easier comparison with the results of their models. Amrhein et al. (2023) call this "Source" system, for example.
>
> 3. You are absolutely right, we are planning a more thorough error analysis as future work. We were not able to report a manual error analysis with a quantification of errors because it can be a very time consuming task, mainly due to the complexity of the morphology of the Portuguese language and the large amount of error types.

---

### Official Review · Reviewer_jbYX · 2023-08-04

**Typos Grammar Style And Presentation Improvements:** 1 . As far as we concern
>> As far a…
**Soundness:** 4

**Excitement:**

3: Ambivalent: It has merits (e.g., it reports state-of-the-art results, the idea is nice), but there are key weaknesses (e.g., it describes incremental work), and it can significantly benefit from another round of revision. However, I won't object to accepting it if my co-reviewers champion it.

**Missing References:**

* Cite Arabic Parallel Gender Corpus (APGC) when your refer to it: https://aclanthology.org/2022.lrec-1.199/
* Please cite the ACL Anthology instead of Arxiv when possible!

**Paper Topic And Main Contributions:**

This paper presents an important contribution in the space of modeling gender-inclusive language, specifically in Portuguese.
The paper created a gold test set and compared two models for the task of gender-inclusive rewriting.
While it may be argued that the task is "artificial" it is still important as current LLM mods will likely suffer from the limited resources in the target form of the language. So, the big picture is timely.

**Questions For The Authors:**

* Why stop only on  25,000 augmented examples to train on?
* Did you try sub-sampling or active learning on the augmented data?


**Reasons To Accept:**

* Well written paper.
* Interesting new contribution - most previous work was Germanic languages; this is on Portuguese, which is quite under represented in NLP.
* New public data set.
* New benchmarks on this task.

**Reasons To Reject:**

* Golden data set is too small (500 examples)
* The annotations used to train the neural model are based on the rule based model... which explains the neural model lower performance than the rule based model.

**Reproducibility:**

4: Could mostly reproduce the results, but there may be some variation because of sample variance or minor variations in their interpretation of the protocol or method.

**Reviewer Confidence:**

5: Positive that my evaluation is correct. I read the paper very carefully and I am very familiar with related work.

---

> ### Author Rebuttal · Authors · 2023-08-28
>
> Thank you for your feedback. If the paper is accepted, we will take into account all your comments.
>
> Regarding your questions and reasons to reject:
> 1. The fact that the neural model was trained on annotations created by the rule-based model is not necessarily the reason why the neural model's performance is lower. This was also the method used by Vanmassenhove et al. (2021), whose neural model achieves a better performance than the rule-based model. Of course, English and Portuguese are languages with very different morphology and in future work we plan on doing a more thorough error analysis.
> 2. We have trained a neural model using 50,000 augmented examples, which achieves similar performance to the one trained with 25,000 examples. Since creating these examples using the rule-based model can be time-consuming and does not entail better results, we have not reported this experiment in the paper.
> 3. We have not sub-sampling or active learning on the data, but we will certainly consider it in future work.

---

### Meta-Review · Area_Chair_YeHC · 2023-09-18

**Recommendation:** 4

**Metareview:**

The work presents a rule-based and a neural-based tool for gender-neutral rewriting for Portuguese and provide the first Portuguese dataset explicitly containing gender-neutral language, including a manually annotated gold standard.

The reviews show ambivalent scores with respect to soundness and excitement, which mostly fall on the upper bound, as it is recognized to be an interesting contribution and topic. Yet, the concerns raised are sensible, and have to do with a fundamental part of the work: the strategy of using a rule-base system to create the data to train the neural system, something which may explain why the rule-based system performs better, even though the authors do not sufficiently discuss this issue. The remaining issues should also be clarified in the final version if accepted (the NER problems, data description, Portuguese examples translated or glossed to facilitate comprehension of gendered terms, etc.)

I should say that a gold standard or 500 sentences for a task like this does not seem too small for a low-resource context. Furthermore, the fact that the paper does not make novel technical (e.g. algorithmic or modelling) contributions should not preclude its publication, as it is  not the work's objective.

---

### Decision · Program_Chairs · 2023-10-07

**Decision:**

Accept-Findings

**Comment:**

The work presents a rule-based and a neural-based tool for gender-neutral rewriting for Portuguese and provide the first Portuguese dataset explicitly containing gender-neutral language, including a manually annotated gold standard.

The reviews show ambivalent scores with respect to soundness and excitement, which mostly fall on the upper bound, as it is recognized to be an interesting contribution and topic. Yet, the concerns raised are sensible, and have to do with a fundamental part of the work: the strategy of using a rule-base system to create the data to train the neural system, something which may explain why the rule-based system performs better, even though the authors do not sufficiently discuss this issue. The remaining issues should also be clarified in the final version if accepted (the NER problems, data description, Portuguese examples translated or glossed to facilitate comprehension of gendered terms, etc.)

I should say that a gold standard or 500 sentences for a task like this does not seem too small for a low-resource context. Furthermore, the fact that the paper does not make novel technical (e.g. algorithmic or modelling) contributions should not preclude its publication, as it is  not the work's objective.